# The Global Water Body Layer from TanDEM-X Interferometric SAR Data

**Jose-Luis Bueso-Bello** *, **Michele Martone**, **Carolina González**, **Francescopaolo Sica**, **Paolo Valdo**, **Philipp Posovszky**, **Andrea Pulella** and **Paola Rizzoli**

Microwaves and Radar Institute, German Aerospace Center (DLR), 82234 Wessling, Germany; michele.martone@dlr.de (M.M.); carolina.gonzalez@dlr.de (C.G.); francescopaolo.sica@dlr.de (F.S.); paolo.valdo@dlr.de (P.V.); Philipp.Posovszky@dlr.de (P.P.); andrea.pulella@dlr.de (A.P.); paola.rizzoli@dlr.de (P.R.)
* Correspondence: jose-luis.bueso-bello@dlr.de

**Abstract:** The interferometric synthetic aperture radar (InSAR) data set, acquired by the TanDEM-X (TerraSAR-X add-on for Digital Elevation Measurement) mission (TDM), represents a unique data source to derive geo-information products at a global scale. The complete Earth's landmasses have been surveyed at least twice during the mission bistatic operation, which started at the end of 2010. Examples of the delivered global products are the TanDEM-X digital elevation model (DEM) (at a final independent posting of 12 m × 12 m) or the TanDEM-X global Forest/Non-Forest (FNF) map. The need for a reliable water product from TanDEM-X data was dictated by the limited accuracy and difficulty of use of the TDX Water Indication Mask (WAM), delivered as by-product of the global DEM, which jeopardizes its use for scientific applications, as well. Similarly as it has been done for the generation of the FNF map; in this work, we utilize the global data set of TanDEM-X quicklook images at 50 m × 50 m resolution, acquired between 2011 and 2016, to derive a new global water body layer (WBL), covering a range from −60° to +90° latitudes. The bistatic interferometric coherence is used as the primary input feature for performing water detection. We classify water surfaces in single TanDEM-X images, by considering the system's geometric configuration and exploiting a watershed-based segmentation algorithm. Subsequently, single overlapping acquisitions are mosaicked together in a two-step logically weighting process to derive the global TDM WBL product, which comprises a binary averaged water/non-water layer as well as a permanent/temporary water indication layer. The accuracy of the new TDM WBL has been assessed over Europe, through a comparison with the Copernicus water and wetness layer, provided by the European Space Agency (ESA), at a 20 m × 20 m resolution. The F-score ranges from 83%, when considering all geocells (of 1° latitudes × 1° longitudes) over Europe, up to 93%, when considering only the geocells with a water content higher than 1%. At global scale, the quality of the product has been evaluated, by intercomparison, with other existing global water maps, resulting in an overall agreement that often exceeds 85% (F-score) when the content in the geocell is higher than 1%. The global TDM WBL presented in this study will be made available to the scientific community for free download and usage.

**Keywords:** water mapping; land cover classification; flood monitoring; bistatic SAR; interferometric coherence; TanDEM-X; InSAR

## 1. Introduction

Covering about 71% of the Earth's surface, water represents the most widespread environment of our planet [1]. It is one of the most precious natural resources on Earth, fulfilling environmental, social, and economic services [2,3]. A reliable and accurate assessment of the world's water resources is, therefore, of paramount importance for a wide range of applications, spanning from resource management, decision making, governance, and climate change initiatives [4–8].

In this scenario, spaceborne remote sensing represents a unique instrument for providing consistent, timely, and high-resolution data on a global scale for water resources mapping and monitoring. Historically, spaceborne optical sensors have been widely used for this purpose [5,9–17], even though they show some limitations caused, for example, by the impossibility to acquire data in the presence of clouds.

Thanks to their all-weather, daylight-independent acquisition capabilities, synthetic aperture radar (SAR) systems represent an attractive alternative to optical imagery [18–21]. Indeed, the use of radar is particularly relevant when tropical regions, mountainous areas, or high-latitudes regions are considered, since they are typically hidden by clouds for considerable time periods, especially during wet seasons or winter time.

To discriminate the presence of water from SAR imagery, amplitude-based algorithms, such as adaptive thresholding, region growing, or basic machine learning classifiers, are widely used [20,22–24]. In particular, given the flatness of water bodies, radar waves are mostly specularly reflected by the surface itself, leading to very low values of backscatter in the acquired image, which are, in some cases, close to the system noise floor. However, the recorded backscatter levels can be influenced by different factors, which make it difficult in certain circumstances to distinguish between water and land. For example, misclassification can occur when illuminating objects with a low radar backscatter, similar to calm water, such as roads and airport runways. On the other hand, the backscatter of water bodies affected by strong wind might significantly increase, depending on the roughness of the illuminated surface, which is strongly influenced by the presence of short-wavelength, shallow-water waves. Moreover, in the presence of deep water waves, characterized by certain regular periodic structures, a coherent superposition of reflections from the faces can occur, leading to the so-called Bragg scattering [25]. In this case, weak individual reflections can sum up to a significant echo signal through constructive interference, leading to high-intensity values within the detected SAR image [26]. Finally, the backscattered signal is also directly influenced by SAR system parameters, such as the local incidence angle and system noise floor (or noise equivalent sigma nought), which can significantly vary on a scene basis.

SAR interferometry (InSAR) allows for overcoming the above-mentioned limitations of backscatter-based algorithms, by introducing the use of the interferometric coherence for water mapping purposes. The interferometric coherence is defined as the normalized cross-correlation coefficient between the interferometric image pair and represents the key quantity for assessing the quality of an interferogram [27]. In InSAR acquisitions, water bodies typically show very low values of coherence and are generally characterized by a more stable behavior, less influenced by surface roughness caused by waves and strong winds. In repeat-pass InSAR systems, the interferometric coherence suffers from temporal decorrelation, which can significantly decrease the coherence for unstable land cover classes, such as forests or agricultural areas, hence making impossible to distinguish such land cover classes from water bodies. This limitation is overcome by the interferometric coherence, obtained from a bistatic system, such as TanDEM-X, where the interferometric pair is acquired simultaneously; therefore, InSAR data are not affected by temporal decorrelation.

In [28], the bistatic interferometric coherence was exploited, together with backscatter, for the derivation of the TanDEM-X water indication mask (WAM), by setting a series of empirical thresholds. The WAM's purpose is to indicate possible water surfaces appearing in the mosaicked global TanDEM-X DEM. Nevertheless, such a product is strongly affected by the presence of misclassified pixels, mainly caused by geometric distortions over high-relief terrain, such as shadow and layover, which characterize the side-looking geometry of SAR and lead to a drop in the interferometric coherence and strong backscatter modifications. Additionally, the WAM is of difficult usage, since it does not consist of a simple binary layer (water/non-water) but counts the number of occurrences of water detection, based on the thresholding of different input observables (such as backscatter, coherence, or a combination of the two) in the mosaicked DEM product. It is then up to the

user to select which case, among the mentioned ones, has to be considered for generating a binary layer, and this choice is far from trivial.

Furthermore, the TanDEM-X DEM product, in the forms of the WorldDEM [29], Copernicus DEM [30], and DLR TanDEM-X DEM [31], currently represents the most accurate and recent reference DEM, at the global scale, for both the scientific and commercial communities [32]. The availability of a reliable water surface product, which is consistent with such DEM data, is, therefore, of paramount importance for a variety of applications, spanning from enhanced earth observation (EO) data processing and hydrological risk assessment to land cover change monitoring and shore line detection. For example, this need was already highlighted during the development of an automatic editing procedure for the global TanDEM-X DEM, as documented in [33].

All these considerations were the main drivers that brought us to the development of a new water body layer from TanDEM-X data, i.e., the TDM WBL, characterized by improved accuracy, with respect to the WAM, which provides ready-to-use information to the end-users. As an additional purpose, the TDM WBL can also contribute to the assessment of water bodies, at a global scale, and monitoring of their evolution in time, by including an estimate of their changes during the past decade (2010–2020).

In this paper, we report on the work developed at the DLR Microwaves and Radar Institute to generate a new global layer to be added to the suite of present TanDEM-X products. We describe the implemented method to detect water bodies from the global data set of TanDEM-X acquisitions, which relies on the use of the bistatic interferometric coherence only. On a scene basis, areas affected by shadow and layover are masked out, to mitigate the misclassification effects of low coherent areas. A watershed segmentation algorithm is then applied to each scene, for the generation of images with contour-closed water regions, which are finally combined through an ad-hoc weighted process to produce the final TDM WBL.

The paper is organized as follows: Section 2 introduces the utilized TanDEM-X interferometric global data set, together with the auxiliary and external data sources required during the generation of the global TDM WBL. The developed method for water classification, based on the watershed segmentation algorithm and a two-step mosaicking strategy, is then described in Section 3. The resulting global water product (TDM WBL), at 50 m × 50 m spatial resolution, is presented in Section 4, together with an accuracy assessment, with respect to selected external reference maps. The findings are then discussed in Section 5; finally, in Section 6, conclusions and outlook are drawn.

## 2. Data

### 2.1. The TanDEM-X Interferometric Global Data Set

TanDEM-X is the first operational spaceborne bistatic SAR system comprising the two twin satellites TerraSAR-X (launched in June 2007) and TanDEM-X (launched in June 2010). The system acts as a large single-pass radar interferometer, nominally acquiring interferometric SAR images in bistatic configuration and stripmap mode (HH polarization), with a typical resolution (azimuth and range) of about 3 m [32,34].

Since the beginning of the mission, more than half a million high-resolution scenes have been acquired and processed for the generation of a global digital elevation model (DEM). A single bistatic scene typically extends over an area of about 30 km, in range by 50 km in azimuth. From this, quicklook images, representing several SAR and InSAR quantities (like backscatter and coherence maps), are generated as a by-product, at a ground resolution of 50 m × 50 m, by applying a spatial averaging process to the corresponding operational TanDEM-X interferometric data at full resolution (12 m × 12 m). Working with such data allows for the exploitation of the TanDEM-X data set at a global scale with a limited computational load and a significant reduction in data volume, memory usage, and processing time [35,36].

The bistatic interferometric coherence gives information about the amount of noise in the interferogram. As described in [37], several error sources may contribute to coherence loss, which, assuming statistical independence, can be factorized as [38]:

$$\gamma_{\text{Tot}} = \gamma_{\text{SNR}} \cdot \gamma_{\text{Quant}} \cdot \gamma_{\text{Amb}} \cdot \gamma_{\text{Rg}} \cdot \gamma_{\text{Az}} \cdot \gamma_{\text{Vol}} \cdot \gamma_{\text{Temp}}. \tag{1}$$

On the right-hand side term of the above equation, $\gamma_{\text{SNR}}$ is the coherence loss due to limited SNR, $\gamma_{\text{Quant}}$ represents the coherence loss due to raw data quantization, $\gamma_{\text{Amb}}$ is the decorrelation due to ambiguities, $\gamma_{\text{Rg}}$ is the noise originating from baseline decorrelation component, $\gamma_{\text{Az}}$ is the contribution caused by the relative shift of Doppler spectra, $\gamma_{\text{Vol}}$ is the decorrelation due to volume scattering effects, and $\gamma_{\text{Temp}}$ represents the temporal decorrelation. As TanDEM-X operates as a single-pass radar interferometer, it is not affected by temporal decorrelation, i.e., $\gamma_{\text{Temp}} = 1$. Differently, $\gamma_{\text{Vol}}$ is not negligible when radar waves penetrate into volumetric targets, such as vegetated areas or snow-covered regions. In this case, the amount of decorrelation is closely related to the height of ambiguity, $h_{amb}$, which represents the topographic height difference, corresponding to a complete $2\pi$ cycle of the interferometric phase [39]. For the bistatic case, it is defined as follows:

$$h_{amb} = \frac{\lambda \cdot R \cdot \sin(\theta_i)}{B_{\perp}}, \tag{2}$$

with $\lambda$ being the radar wavelength, $R$ the slant range, $\theta_i$ the incidence angle, and $B_{\perp}$ the baseline perpendicular to the line-of-sight. In this work, we make use of quicklook images, representing both the interferometric coherence and height of ambiguity as main observables for the detection of water bodies in TanDEM-X InSAR acquisitions, as well as for deriving a reliable mosaicking strategy to generate a new global WBL product.

### 2.2. Auxiliary Data

Different auxiliary data sets have been used to enhance the TDM WBL generation algorithm and validate the final product. To improve the classification and subsequent combination and mosaicking process, both internal (i.e., obtained from the TanDEM-X products and annotation) and external information sources have been used during the generation of the TDM WBL, which are detailed in the following.

All the used internal data sets, derived from TanDEM-X data and acquisition parameters, are listed in the following:

- Shadow and layover mask (SLM): Geometric distortions, such as shadow and layover, are observed in SAR images as low-coherence areas. This effect mainly occurs over mountainous terrain and may lead to the wrong classification of water bodies, when using approaches based on coherence thresholding [40]. For each considered scene, we detected such areas by applying the approach proposed in [41], which takes into account the properties of each SAR scene and its acquisition geometry (orbit height, baseline, and incidence angle). By combining such information with an external reference DEM (in this case, the edited version of SRTM DEM [18], detailed later on), it is possible to detect areas with low coherence on the SAR image, which corresponds to geometric distortions, namely the shadow and layover regions. Figure 1 shows an example of the shadow and layover map, derived for a TanDEM-X image over the Alps, Europe. Amplitude and coherence images are presented as a reference, together with the obtained shadow and layover map. Additionally, a map of the local slope is generated, as well, by computing the bi-dimensional gradient of the reference DEM.
- TanDEM-X quality check products: TanDEM-X acquisitions are interferometrically processed by the operational Integrated TanDEM-X Processor (ITP) [42]. During all the processing chain, the ITP provides direct feedback on the acquisition quality, to ensure high performance of the produced single-scene DEMs [32]. Remaining errors, which may contribute to possibly larger DEM errors, are phase unwrapping and on-board oscillators synchronization problems [43], which are annotated in the ITP quality check

products. Moreover, the generated TanDEM-X single scene DEMs by the ITP are then combined in the subsequent TanDEM-X DEM mosaicking and calibration processor (MCP) [44]. Residual errors, due to the remaining phase unwrapping problems or presence of heavy-raining clouds, are annotated in the MCP quality check products. Both ITP and MCP quality check products are considered in the weighted mosaicking process for the generation of the TDM WBL, as well.

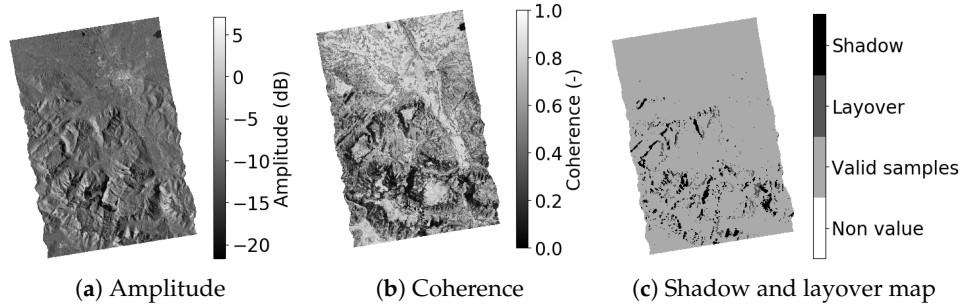

**Figure 1.** Example of a TanDEM-X image over the Alps, close to Königsee, Germany. (**a**) Amplitude, (**b**) interferometric coherence, and (**c**) derived shadow and layover map, to mitigate missclassification of water bodies over mountainous regions.

Several auxiliary external data sets have also been utilized during the generation of the TDM WBL to improve the water classification and mosaicking strategy.

- SRTM (Shuttle Radar Topography Mission, [45]): An enhanced, edited version of the SRTM DEM has been used as reference for the detection of shadow and layover regions in each TanDEM-X image. The used SRTM DEM has been merged from an edited SRTM C-Band DEM, adjusted with ICESat, and the Global Land One-km Base Elevation (GLOBE) DEM, as described in [46].
- MODIS (moderate resolution imaging spectroradiometer, [47]): The global snow and ice map, provided monthly by MODIS, at a spatial resolution up to 500 m, is used to detect TanDEM-X scenes affected by the presence of snow, which might lead to a misclassification of water surfaces.
- OSM (OpenStreetMap, [48]): OSM provides a global skeleton of rivers, which are used as input for the watershed classification algorithm, to enhance the placement of the user-defined water markers at the resolution of TanDEM-X quicklooks.
- GlobCover [49]: The backscatter values over sandy desert regions are often close to the TanDEM-X SAR system sensitivity and can lead to an incorrectly estimated low interferometric coherence. In order to avoid the misclassification of such areas as water bodies, the GlobCover classification map has been used to mask out desert regions, as already done for the global TanDEM-X forest/non-forest map [36].

For the validation and performance assessment of the TDM WBL we utilized the following independent data sets:

- Copernicus water and wetness (WAW) layer [50]. The Copernicus WAW layer, available over Europe, has been used for the validation of the TDM WBL. The WAW layer is part of the pan-European high-resolution layers (HRL), which provide information on specific land cover characteristics at a 20 m × 20 m resolution.
- TanDEM-X WAM Layer (TDM WAM) [51]. The TDM WAM is delivered together with the TDM global DEM and has been generated during the mosaicking of the full-resolution TanDEM-X DEM at 12 m × 12 m. It is an occurrence counter mask based on the thresholding of both the amplitude and the interferometric coherence. In particular, two fixed thresholds for all acquisitions have been defined for the amplitude: a relaxed amplitude threshold of −15 dB and strict amplitude threshold of −18 dB, while the threshold applied to the interferometric coherence is 0.23. For each single TDM image used for the generation of the global DEM, image pixels showing values below these

thresholds are flagged as water. Then, during the mosaicking process, and on a geocell basis, the total occurrence of detected water from overlapping scenes is evaluated and coded into an 8-bit map. In other words, for each pixel in the WAM a coded value is saved, which reflects the number of overlapping acquisitions under the specified thresholds up to a maximum of 3 occurrences. The complete description of the WAM bits coding can be found in [51]. In order to convert the multiple bit coding of the WAM into binary layers, we split the WAM information into different categories, leading to the generation of up to 13 different binary water maps. All possible combinations and the relative products are summarized in Table 1. For each generated water map, pixels coded with other bits combinations in the WAM have been considered as invalids. Note that the column "All counters" means that, at least in one acquisition, water has been detected for the corresponding WAM binary layer.

**Table 1.** Amplitude and coherence pixels combinations used for generating the 13 binary water/non-water layers from the WAM. Each row displays a different input information: coherence and amplitude (Coh + Amp), coherence only (coherence), amplitude only (using the −15 dB relaxed threshold (Amp. < −15 dB)), and amplitude only (using the more stringent threshold of −18 dB). Each column identifies the associated number of occurrences in water detection ("Acq. counter"). The column "All counters" corresponds to the union of the last three columns. Cells marked with an x identify the generated products.

| WAM Binary Layer | All Counters | Acq. Counter | | |
|---|---|---|---|---|
| | | 3 | 2 | 1 |
| Coh + Amp | x | - | - | - |
| Coherence | x | x | x | x |
| Amp. < −15 dB | x | x | x | x |
| Amp. < −18 dB | x | x | x | x |

- ESA CCI water map [6]: The freely available global map of open permanent water bodies obtained from the Land Cover (LC) project of the Climate Change Initiative (CCI), provided by ESA, at 150 m × 150 m resolution, is used for a large-scale inter-comparison of the produced water maps.Z.
- FROM-GLC water map [14]: The FROM-GLC (Finer Resolution Observation and Monitoring of Global Land Cover) water map has been used for the large-scale inter-comparison of the TDM WBL. This water map has been generated using a machine learning random forests classifier, trained on Landsat data, and updated to 2017 using additional Sentinel-2 data. It is more up-to-date than the ESA CCI one and has been generated at a resolution of 10 m.
- GSW occurrence map [5]: The GSW (global surface water) occurrence map from the European Commission (EC) Joint Research Centre (JRC) is based on Landsat imagery. It shows, at 30 m resolution, the frequency with which water was detected on the surface, from 1984 up to 2015, at the global scale. In order to generate a binary layer to be used for the comparison with the TDM WBL, we set an empirical threshold at the 50% water occurrence.

Finally, it is worth pointing out that, before usage, all the described data sets are interpolated to the corresponding latitude/longitude grid and pixel size of the considered TDM input/output product.

## 3. Methods

The main processing steps of the developed algorithm, for the generation of the TDM WBL, are illustrated in Figure 2, where three major blocks can be distinguished: (I) data preparation, applied in order to select and derive the required input features for water

detection from TanDEM-X original quicklooks, (II) seeds placement, for the subsequent application of the detection algorithm, and (III) classification + mosaicking of the final product. The last block comprises of the classification of single images, setting of reliable weights for the mosaicking process, and combination of the overlapping scenes into the final mosaic. The different blocks are separately described in Sections 3.1–3.5. Finally, Section 3.7 describes the parameters used for the final accuracy assessment.

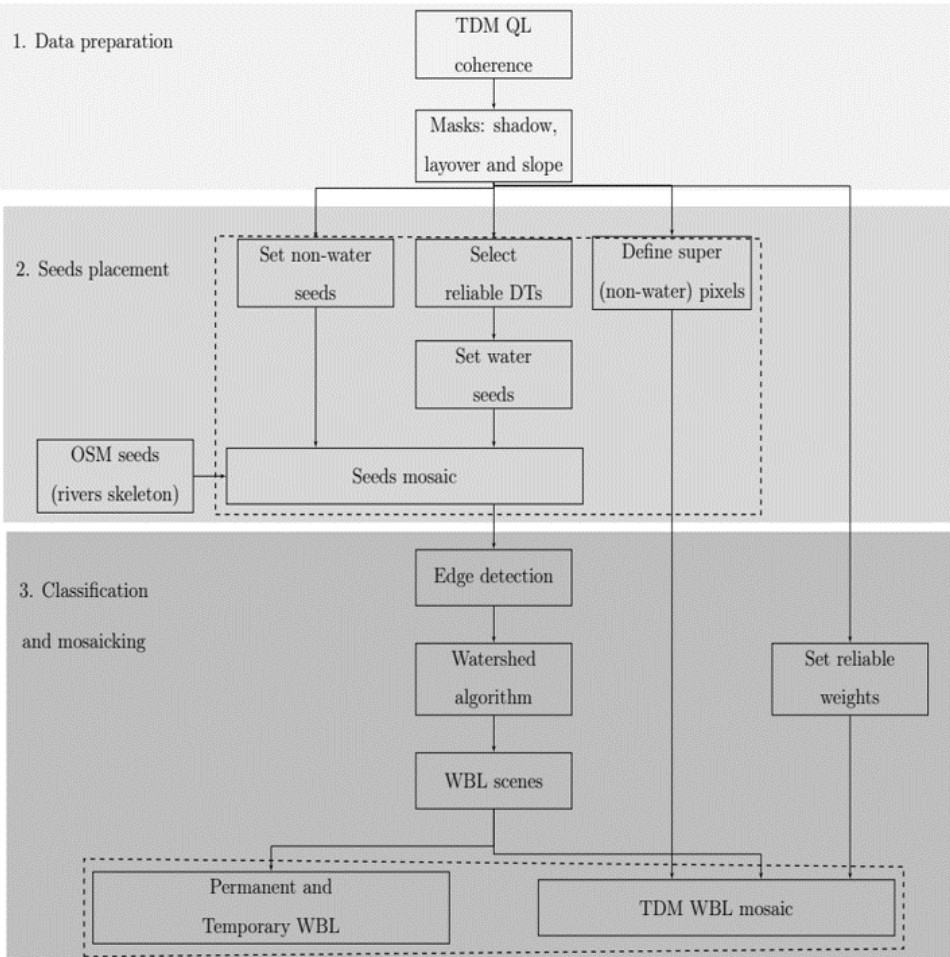

**Figure 2.** TDM WBL method flowchart. Solid line rectangles indicate operations at scene basis. Dashed rectangles indicate operations at mosaic level. Grey filled blocks indicates the three major blocks of our algorithm.

### 3.1. Data Preparation

The TanDEM-X interferometric coherence images are used as input for the water body classification algorithm. The data preparation comprises of the masking of shadow and layover regions, filtering of high slope terrain, and extraction of auxiliary information. All the mentioned operations are performed on each single image, separately. In particular, pixels affected by shadow and layover are set as invalid. Moreover, because of the topography of high-slope regions, lakes cannot be found on high slopes. By utilizing the local slope map, pixels whose slope exceeds the empirical value of 10° are set as non-water pixels on the single TanDEM-X coherence images.

The auxiliary information on the identification of problematic scenes is extracted from different sources, as described in Section 2.2, and is used in the next steps of the algorithm for setting both the seeds for the classification algorithm and weights for the final mosaicking process.

### 3.2. Seeds Placement

The second step in the TDM WBL classification algorithm is the placement of appropriate seeds, prior to the application of the watershed segmentation algorithm. Seeds are defined as the catchment basin on a topographic surface. As shown in Figure 3, considering this surface the height of each point in the map, if some water is dropped on it, the water will stream down, reaching a minimum height and stopping there. All points of the surface in which the drops of water reach this minimum are called seeds [52]. They represent the starting points for the location of water bodies in our classification algorithm.

The seeds placements procedure takes into account the following aspects:

- Reliable data takes: In the proposed method, data takes that are affected by snow and clouds, showing an interferometric low quality, or with a height of ambiguity lower than 25 m are considered as non-reliable acquisitions and are not used further. By excluding data takes affected by the presence of snow, the probability to correctly set water seeds over seasonally frozen lakes increases, thanks to a more likely usage of summer acquisitions, if available. Regarding data takes acquired with low height of ambiguities (or, alternatively, large normal baselines), they typically show low coherence values over forested areas because of the high impact of volume decorrelation, which can mislead the classification [39].

- Water and non-water seeds: Once the reliable data takes have been selected, we define seeds for both water and non-water bodies by properly thresholding the input coherence. The reference threshold values have been empirically defined, after a statistical analysis of more than 200 TanDEM-X images, acquired using different geometries and acquisition parameters. By comparing these images with the ESA CCI water map [6], it has been possible to statistically characterize the expected coherence values for water and non-water bodies. For water bodies, a coherence reference value of 0.22 has been obtained, similar to the one employed by [28], and relatively close to the lower coherence bias. For non-water bodies, the coherence value depends on the land cover type under evaluation. A coherence reference minimum value of 0.5 has been selected as representative for all land cover types. These coherence reference values are the input parameters for the watershed algorithm.

- *Super pixels*: For a given pixel location, in case all available coherence images from overlapping multiple acquisitions show a coherence value above 0.6, this pixel is directly set as non-water in the final mosaic, since persistently high coherence values are a reliable indicator of the absence of water [38].

- OSM rivers skeleton: Working with a pixel resolution of 50 m $\times$ 50 m on ground, narrow river beds smaller than a pixel cell are challenging to detect. The backscattered signal of the surrounding land is merged with the response of such small water regions, and the obtained coherence is higher than the expected one for pure water bodies. This effect leads to a difficult positioning of water seeds. In order to correctly detect such water bodies, we complement seeds detection using the Open Street Map (OSM), which provides a global skeleton of rivers [48]. We extracted this information using the OSMxtract Python package [53]. The skeletons of narrow rivers are tagged as waterways in the OSM and such coordinates on ground are set as water seeds.

- Seeds mosaic: Finally, for each single output coordinate on ground (latitude $\times$ longitude), we consider all the $N$ available overlapping acquisitions simultaneously. If it holds that:

$$\frac{\sum_i^N \text{Water seeds}(i)}{\sum_i^N \text{Reliable data takes}(i)} > 0.4, \tag{3}$$

such a pixel is set as water seed in the final seeds mosaic. Note that a relaxed threshold has been considered here, since only reliable data takes are taken into account in this first mosaicking process. The result is a mosaic of seeds that contains water and non-water seeds, as well as super pixels, which is then used as input for the next algorithm steps.

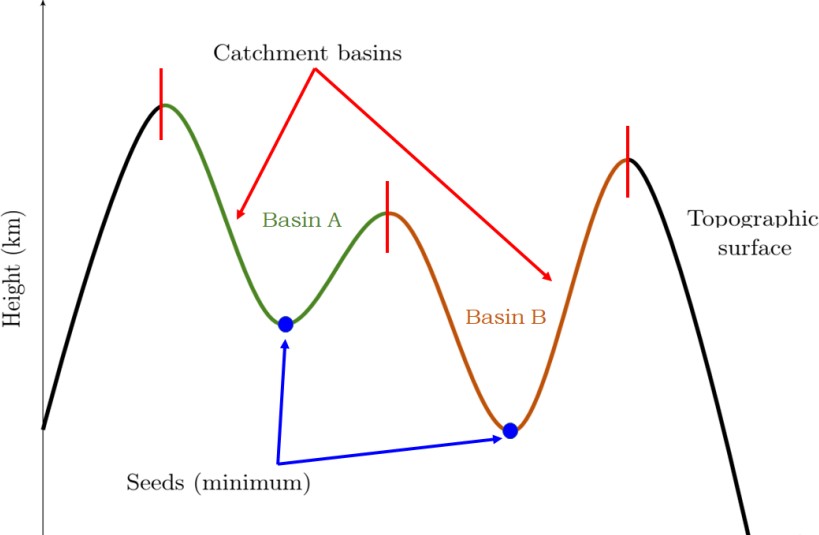

**Figure 3.** Principle of water seeds placement for the watershed algorithm. They are identified in blue as the topographic minimum of catchment basins.

### 3.3. Single-Scene Water Classification

For the generation of the global TDM WBL, we developed a classification method based on the watershed segmentation algorithm [52]. The watershed algorithm is a well-known and commonly used approach in image segmentation, such as bubble detection in a grey scale image. It is a non-parametric contour detection method, with the advantage that no empirical threshold values are needed. The output are closed contour lines that delineate the borders between regions with different characteristics. Starting from the set of predefined seeds, the watershed algorithm treats input pixels values as a local topography (elevation). Each basin is ideally flooded in an iterative way, and the areas where the flood-waters from different basins meet are identified as barrier contours. Such contours represent the different partitions in the image, which can, in this way, be properly segmented.

In the method presented in this paper, we rely on the implementation of the watershed algorithm, provided by the scikit-image package for Python [54], using the watershed-by-flooding approach. As markers for the watershed algorithm, the defined seeds mosaics in Section 3.2 are used. The topographic representation of the input TanDEM-X coherence images is obtained by applying a Scharr transform, which is a filtering method used to identify and highlight gradient edges and features by applying a bi-dimensional kernel, representing the first derivatives [55]. The watershed algorithm is then applied on such images, starting from the defined seeds. The resulting contoured regions where the water seeds were placed are identified as water bodies.

### 3.4. Reliability Weights

When dealing with a large set of data, such as the one provided by TanDEM-X, it is necessary to define an appropriate way to combine together the overlapping scenes, e.g., by giving precedence to more reliable acquisitions, with respect to poor-quality ones. For each pixel in the output mosaic, we derive its value (i.e., water/non-water) by applying a weighted average to all the $N$ input pixels from overlapping scenes. The correct definition of the mosaicking weights $\alpha_i$ plays a key role for the enhancement of the final classification performance [36]. In the case of the TDM WBL, a single weight value is defined for each input image, which is established on the basis of both annotated parameters and derived quantities from the acquisition geometry and interferometric processing quality check products. For a certain output pixel, given a set of $N$ input overlapping images, a

unit weight $\alpha_i$, where $i = [1, \cdots, N]$ is associated with each of them, $\alpha_i$ results from the multiplication of different terms, which are initially set equal to 1, as:

$$\alpha_i = \alpha_i^{snow} \cdot \alpha_i^{clouds} \cdot \alpha_i^{acq} \cdot \alpha_i^{winter} \cdot \alpha_i^{h_{amb}}. \tag{4}$$

Each term on the right-hand side of the equation identifies a different source of uncertainty, as explained in the following, and assumes a value that reflects the impact of each error source on the final quality:

- $\alpha_i^{snow}$ quantifies the reliability loss caused by the presence of snow on ground. The coherence over areas covered by dry snow is typically degraded because of volume decorrelation effects, while, in the presence of wet snow and bare ice, such a phenomenon is negligible at the X band [56]. The snow coverage information is obtained from MODIS [47]. If the percentage of snow, indicated by MODIS over an image, is higher than the empirical threshold of 20%, then this is considered as a moderate source of uncertainty and $\alpha_i^{snow} = 0.5$.

- $\alpha_i^{clouds}$ characterizes acquisitions affected by heavy-rain clouds, which appear in the coherence images as low coherent areas and could be identified as water surfaces. The clouds information is obtained from the MCP quality check products, described in Section 2.2. If heavy rain events are detected, they are considered as critical error sources and $\alpha_i^{clouds} = 0.1$.

- $\alpha_i^{acq}$ is associated to the presence of acquisition problems, eventually annotated in the ITP quality check products, introduced in Section 2.2. Additionally, in this case, if anomalies are reported, $\alpha_i^{acq} = 0.1$.

- $\alpha_i^{winter}$ quantifies the reliability of winter acquisitions. Specifically, during this season, water bodies without constantly flowing water, such as lakes, may be frozen. This condition changes their backscattering properties, and they appear as more coherent areas. In this case, $\alpha_i^{winter} = 0.5$. One should note that we define as winter the time period between October and April for data takes acquired over regions at latitudes higher than 30°N and between April and October for data takes acquired over regions at latitudes lower than 30°S.

- $\alpha_i^{amb}$ accounts for the interferometric coherence variability, with respect to the height of ambiguity $h_{amb}$ [38]. For low values of $h_{amb}$, the coherence over forested areas can be degraded to values close to the lower bias [39]. On the contrary, this effect is significantly mitigated with increasing $h_{amb}$, which corresponds to smaller perpendicular baselines. Therefore, we set a different $\alpha_i^{amb}$ value, depending on specific $h_{amb}$ intervals and seasonal time, as summarized in Table 2.

**Table 2.** Reliability weights $\alpha_i^{amb}$ applied in the TDM WBL as a function of the height of ambiguity $h_{amb}$ for summer and winter data takes (DT).

| Height of Ambiguity | Summer DT | Winter DT |
|:---:|:---:|:---:|
| <40 m | 0.5 | 0.5 |
| >60 m | 2.0 | 0.5 |
| >80 m | 4.0 | 1.0 |

### 3.5. Final Mosaicking

The final TDM WBL mosaic is provided on a basis of $1° \times 1°$ geocells, similarly to the global TanDEM-X DEM product [32] and TanDEM-X forest/non-forest map [36]. In order to generate the final mosaic, a three-dimensional (3D) data cube for each output geocell is first created by stacking the input scene-based binary water images on a common latitude ($\phi$) and longitude ($\lambda$) grid. For each ($\phi, \lambda$) coordinate, all available overlapping images,

together with the associated reliability weights, are considered, and the probability to be water $W_{mosaic}(\phi, \lambda)$ is then computed as:

$$W_{mosaic}(\phi, \lambda) = \frac{\sum_i^N \alpha_i(\phi, \lambda)\, w_i(\phi, \lambda)}{\sum_i^N \alpha_i(\phi, \lambda)}, \tag{5}$$

where $w_i(\phi, \lambda)$ indicates the binary water map value from a single image, resulting from the application of the watershed classification algorithm. The final binary classification of water/non-water is obtained after setting a threshold at 35% on $W_{mosaic}(\phi, \lambda)$. Pixel values above this threshold are selected as water in the final TDM WBL mosaic. Such a threshold value has been empirically set after an experimental analysis on the final accuracy.

Together with the TDM WBL, an auxiliary map is generated, which gives indications on the presence of permanent and temporary water bodies. This mosaic is obtained by simply counting the number of water occurrences within the three-dimensional data cube of overlapping images. For each water body, the area identified as water in all overlapping images is classified as permanent water. Other areas, identified as water only in some of the available images, are classified as temporary water. This map represents an additional valuable information for the end-users, since it is a good indicator for tracing temporal changes. Specifically, it shows changes in the extension of water bodies as measured by TanDEM-X during the period 2011–2016, capturing, for example, the seasonality of frozen lakes during winter or water accumulations during the melting of snow packs in mountainous regions. Moreover, it records the variations of water riverbeds and lakes, which account for eventual flooding events or seasonal changes in water reservoirs.

### 3.5.1. Frozen Water

The detection of frozen water surfaces represents a challenging aspect in the framework of water mapping and deserves to be specifically addressed. Regions close to the Arctic circle are characterized by the presence of frozen water during most of the year, which results in higher coherence values within InSAR acquisitions. In order to achieve global coverage with TanDEM-X data over such regions, we had to face the problem of having areas where only winter acquisitions are available. This aspect leads to a high probability of misdetection of water bodies. As an example, Figure 4 shows two mosaics obtained by superimposing all the available TanDEM-X coherence images over the Yana Bay, north-western Russia, from 2011 to 2016. This is a coastal region facing the Arctic ocean. Figure 4a,b depict the coherence mosaics, obtained by considering TanDEM-X summer and winter acquisitions separately.

The area faces the Arctic Ocean to the north and is characterized by the presence of small lakes in the inland region and the Yana river mouth to the east. The images composing the summer mosaic (Figure 4a) show a high variability in ground conditions. The ocean appears with the expected low coherence in just one image (the second one from the left-hand side). All other images present higher values than expected over water, indicating the presence of sea ice even during summer time. The small lakes in the inland region mainly appear as low coherent areas, even though some images, as for the ocean, still show frozen lakes, characterized by very high coherence. On the other hand, in the mosaic of winter acquisitions (Figure 4b), high coherence values are obtained for all water bodies, as well as over land, indicating the presence of frozen water surfaces and snow on the ground, respectively.

The previous observations have also been verified by statistically analyzing single TanDEM-X coherence images over the Yana Bay test region, separately. Water bodies and land areas have been differentiated on the basis of the ESA CCI water map [6]. Exemplary results for two images, acquired in winter and summer time, respectively, are provided in Figure 5. For each acquisition, a masked coherence image for water bodies and land areas, as well as the corresponding histograms of the respective coherence values, are depicted. Figure 5a shows an image acquired in winter 2011, close to the Arctic ocean. Here, similar coherence values are obtained for non-water and water bodies, since the latter

are frozen. The high level of coherence over the ocean is caused by the presence of sea ice. The latter is typically characterized by a significant surface roughness, which leads to much higher backscattering returns toward the radar antenna, with respect to open water, where specular reflections occur. Differently, Figure 5b depicts an image acquired during summer 2012. The Arctic ocean is now free from sea ice and, as expected, shows very low coherence values. It can also be noted that some small lakes remain frozen and present coherence values, similar to the non-water ones. Therefore, this analysis confirms how coastal areas at northern latitudes are mainly distinguishable in summer acquisitions only, when no sea ice is present. Nevertheless, even at such conditions, the detection of remaining frozen surfaces is still challenging.

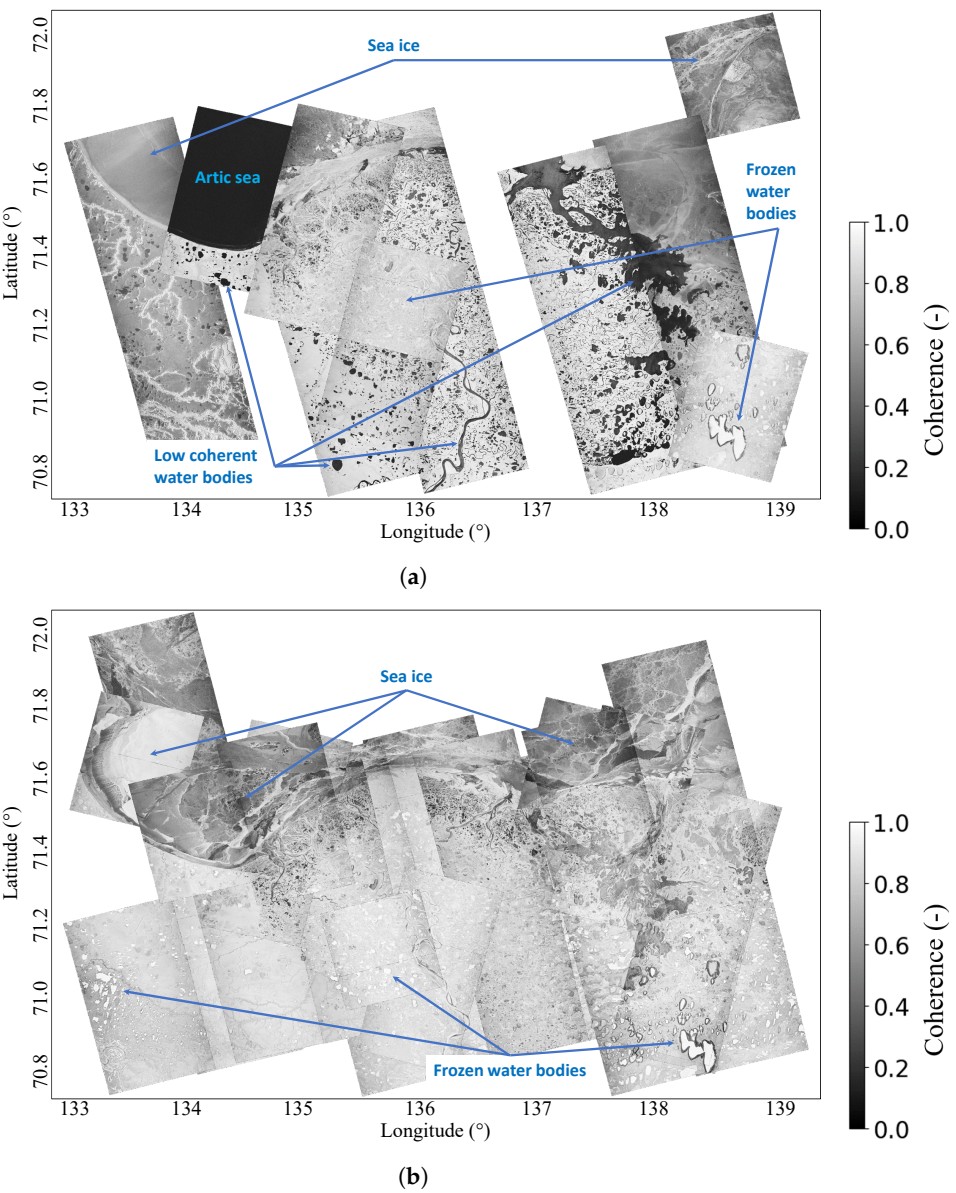

**Figure 4.** Mosaics of TanDEM-X coherence images, considering summer (**a**) and winter (**b**) acquisitions, separately, over a test area close to the Yana Bay (Russia), facing the Arctic Sea to the north.

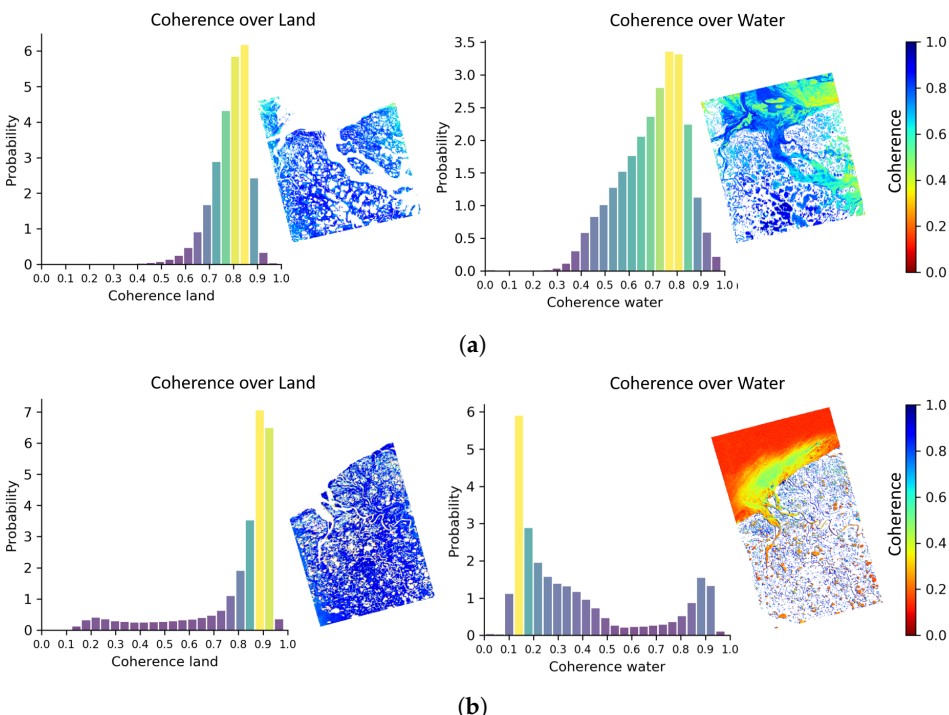

**Figure 5.** TanDEM-X coherence images and histograms for water bodies and land areas over the Yana Bay (Russia) test area. Both winter (**a**) and summer (**b**) acquisition cross the coastline of the Arctic Ocean. (**a**) TanDEM-X acquisition from January 2011. (**b**) TanDEM-X acquisition from July 2012.

For all these reasons, and in order to mitigate the effects of these extremely varying conditions, an ad-hoc approach has been applied when mosaicking regions at latitudes higher than 60°N. We added an additional layer to our final TDM WBL mosaic: frozen water, detected as high coherent areas with $\gamma_{\text{tot}} > 0.95$ in data takes acquired during winter. Note that this behaviour is only observable for inland water bodies in a reliable manner. For coastal regions, as it is the case, e.g., of geocells over Greenland, it has been necessary to adapt the reference coherence values for the seeds placement (Section 3.2). In this case, a reference value of $\gamma_{\text{tot}} < 0.55$ has been set for water seeds, while for land areas, mainly covered by snow and ice, a reference value of $\gamma_{\text{tot}} > 0.7$ has been applied. The modified threshold values have been empirically selected after the statistical analysis of different test images such as the ones presented in Figure 5. Moreover, super pixels have not been considered for the generation of the TDM WBL over these regions, and that at coastal areas, when only winter data takes were available; the shoreline, corresponding to the open ocean, has been extracted from the ESA CCI water map [6].

### 3.6. Additional Output Information Layers

Together with both described water maps, i.e., the TDM WBL and the temporary and permanent WBL, two additional information layers are generated after the mosaicking process:

- Coverage map (CM): A map indicating the number of mosaicked acquisitions for each latitude/longitude pixel coordinate of the TDM WBL.
- Acquisition information files (AIF): The acquisition information files list all the acquisitions used in the generation of the TDM WBL map on a geocell level. The list contains the data take acquisition identifier, its scene number, and the date of the acquisition.

### 3.7. Accuracy Assessment

The quality of the produced TDM WBL has been assessed by computing the confusion matrices, with respect to external reference maps, and evaluating several quality parameters. In the case of the TDM WBL, two classes are considered, water and non-water, as indicated in Table 3.

**Table 3.** Confusion matrix for TDM WBL accuracy assestment.

| | | Reference Map | |
|---|---|---|---|
| | | **Water** | **Non-Water** |
| TDM | Water | *TP* | *FP* |
| WBL | Non-water | *FN* | *TN* |

The four terms in the confusion matrix are defined as: true positives (*TP*): pixels classified as water in both maps; false positives (*FP*): pixels classified as water in the TDM WBL and non-water in the reference map; false negatives (*FN*): pixels classified as non-water in the TDM WBL and as water in the reference map; and true negatives (*TN*): pixels classified as non-water in both maps. The structure of the confusion matrix is depicted in Table 3. Starting from such a matrix, several widespread metrics, commonly used for the accuracy assessment of land classification algorithms, can be derived. For the quality assessment of the TDM WBL, overall accuracy (OA), F-score, and Matthews correlation coefficient (MCC) are considered [57]. In the following, we briefly recall their definitions and comment on their choices. In particular:

- The overall accuracy (OA) represents the overall correctly classified pixels, with respect to the total number of classified pixels, and is defined as:

$$OA = \frac{TP + TN}{TP + TN + FP + FN} \qquad (6)$$

 The OA is provided for completeness, since it is well known that it shows optimistic results, especially on imbalanced data sets. Indeed, in the case of water mapping at a global scale, the proportion of water is often marginal, with respect to the non-water class, as shown in [6].

- F-score, also called the F1-score, is an accuracy metric that ranges between 0 and 1 and can be expressed as:

$$F - score = \frac{2 \cdot TP}{2 \cdot TP + FP + FN} \qquad (7)$$

 F-score is mainly used to evaluate binary classifications, and it is specially useful when dealing with imbalanced data sets. The overall accuracy in Equation (6) has the advantage to be easily interpretable, but the disadvantage is that it is not very robust when the data is unevenly distributed. The F-score metric represents a useful alternative when dealing with such kind of data sets.

- The Matthews correlation coefficient (*MCC*) measures the statistical relationship between classified and reference classes and is defined as:

$$MCC = \frac{(TP \cdot TN) - (FP \cdot FN)}{\sqrt{(TP + FP) \cdot (TP + FN) \cdot (TN + FP) \cdot (TN + FN)}} \qquad (8)$$

 The *MCC* is often used to assess the quality of binary classification, since it is generally regarded as a balanced measure of accuracy, even in the presence of classes with very different population sizes [57,58]. The MCC index varies between $-1$ and 1. $MCC = 1$ represents a perfect agreement between the classification and reference maps. $MCC = 0$ means that the classification approach is no better than a random prediction approach. $MCC = -1$ indicates an absolute disagreement between

classification and reference maps. The *MCC* assumes a high score, only if a good classification is obtained in all four terms of the confusion matrix.

## 4. Results

### *4.1. The Global TanDEM-X Water Body Layer*

The final TDM WBL map, with a spatial resolution of 50 m × 50 m, is shown in Figure 6a. More than 500,000 TanDEM-X bistatic scenes, acquired between 2011 and 2016, have been processed and mosaicked for its generation. The map is divided into geocells of 1° × 1° in latitude and longitude, as done for the production of the TanDEM-X global DEM. Water bodies are depicted in blue and non-water pixels or land pixels are depicted in white. Moreover, even if not visible at this zoom-in level, invalid pixels, where no-data is available, are shown in black; shadow and layover pixels are shown in red. During the TanDEM-X mission, the complete Earth's landmasses, i.e., all land areas, such as a continent that is in one piece and not broken up by oceans, have been covered at least twice. Therefore, areas where no TanDEM-X scenes are available have been considered as open water, corresponding to oceans, and are depicted in blue in Figure 6a, as well.

In addition to the binary TDM WBL map, the TDM permanent and temporary WBL is produced, as well, and is presented in Figure 6b. This map has the same spatial resolution and geocell division as the binary TDM WBL. The temporary WBL depicts water bodies variations, such as flooded regions (due to the swelling of rivers), lakes shrinking (due to periods of drought), or frozen water surfaces during winter. Such effects are clearly visible, e.g., in the northern regions of the boreal hemisphere, corresponding to permafrost regions, where water completely freezes during winter, or in wetlands areas close to the major tropical rivers.

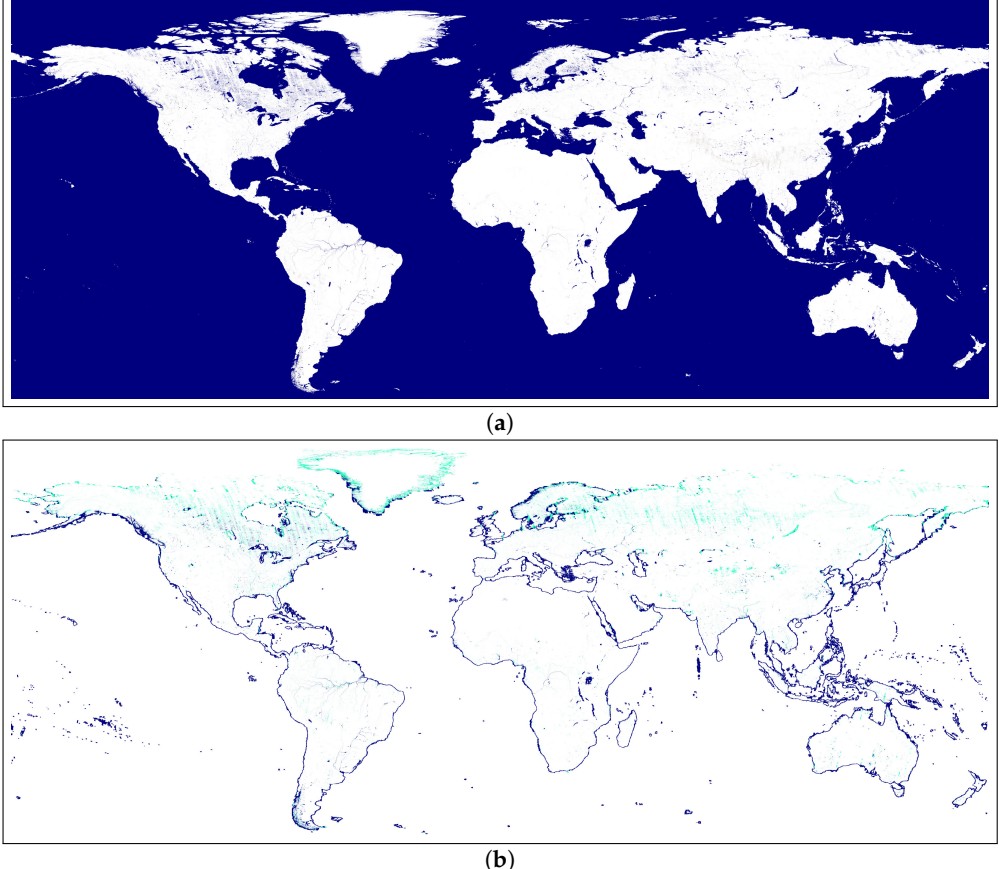

(**a**)

(**b**)

**Figure 6.** (**a**) The Global TanDEM-X water body layer (blue: water surfaces, white: landmasses). (**b**) The global permanent and temporary WBL (Blue: permanent water bodies, cyan: temporary water).

A detail of the TDM WBL and the permanent and temporary WBL is shown in Figure 7, depicting the gulf of Martaban in the southern part of Myanmar. This region depicts the Ayeyarwady river, crossing the country from north to south, and its tributaries. The Ayeyarwady river is a wide river, and it shows no significant variation in the width of its riverbed. On the other hand, the tributaries are narrow rivers, which are more difficult to be correctly detected, due to the resolution of the TanDEM-X quicklook images (50 m × 50 m). Nevertheless, thanks to the developed mosaicking strategy, they are well detected, as well (Figure 7a). Moreover, it is worth noting that, depending on the water flow, some regions were temporary flooded during the considered time span for the generation of the TDM WBL. Such effects are clearly caught by the temporary and permanent water layer (Figure 7b).

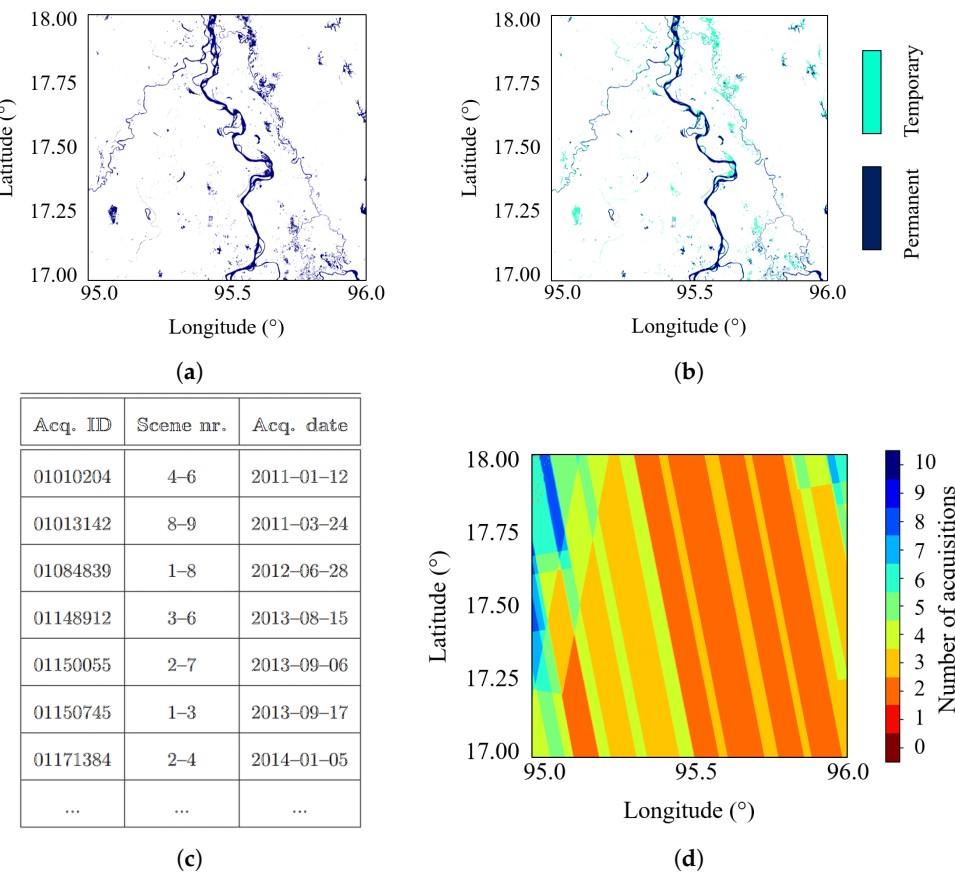

**Figure 7.** Details of the TDM WBL product over the Ayeyarwady River and the Gulf of Martaban in the South of Myanmar. (**a**) The binary mosaicked WBL (water is depicted in blue), (**b**) the corresponding permanent and temporary WBL layer. (**c**) associated acquisition information file (AIF). The AIF content shows the acquisition unique identifier (Acq. ID), the scene number (Scene nr.), and the acquisition date (Acq. date) of the mosaicked scenes at a geocell level. (**d**) Corresponding coverage map (CM).

The corresponding additional output layers are then presented in Figure 7c,d. Figure 7c shows the structure of the acquisition information file (AIF), which lists the data take acquisition identifier, the scene number, and the acquisition date of all used TanDEM-X data takes for the generation of the TDM WBL map on a geocell level. Such an information can be very useful for identifying the exact time span in which variations in the water surface extent are observed in the temporary and permanent WBL. Figure 7d depicts the Coverage Map (CM) associated to the TDM WBL image presented, in Figure 7a. As one can notice, more than 5 overlapping acquisitions were acquired over this area, in both ascending and descending orbit directions.

The examples in Figure 8 allow for additional considerations on the effects of geometric distortions and frozen water in the final product. In particular, Figure 8a shows the TDM WBL over the Tibetan plateau. The corresponding optical image from Google Earth is visible in Figure 8b. Over mountainous regions, the identification of water bodies is quite challenging, due to the strong presence of geometric distortions in SAR images. As it can be seen, these effects have been significantly mitigated by the accurate detection and filtering of shadow and layover in each input TanDEM-X scene. This is particularly visible at the bottom of the image, in correspondence of the Himalayan mountain ridge.

Figure 8c shows an example of frozen water detection for an area in northern Siberia (the corresponding optical image from Google Earth is depicted in Figure 8d). Here, water bodies detected using the TDM WBL watershed classification approach over summer acquisitions are colored in blue. For example, the blue stripe in the middle of the scene corresponds to a TanDEM-X acquisition performed in July 2013. The other ones, detected as frozen water over winter acquisitions, are identified as temporary water and are depicted in cyan.

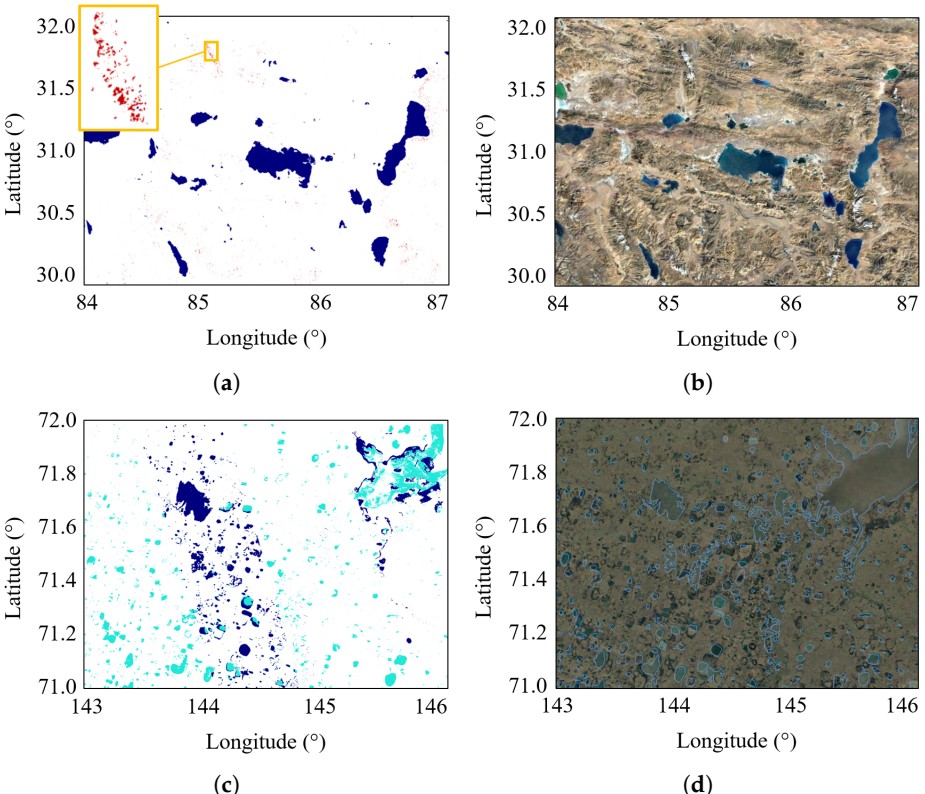

**Figure 8.** (**a**,**b**) Water bodies detected over the Tibetan Plateau, located between the Himalaya mountains on the south and the Taklamakan desert on the north ((**a**) TDM WBL and (**b**) corresponding optical image from Google Earth. In (**a**), water bodies are depicted in blue, other land cover types in white, and remaining pixels with no-data information, due to the shadow and layover mask, are identified in red. (**c**,**d**) Water bodies detection over a test area in Siberia. (**c**) Water surfaces, detected using the watershed classification method, are depicted in blue, while frozen water (water detected in winter acquisitions only) is in cyan. (**d**) Corresponding optical image from Google Earth.

## 4.2. Accuracy Assessment

In this section, the global TDM WBL is validated with external references and compared with other available large-scale classification maps. In order to perform a reliable validation, it is necessary to dispose of a highly accurate reference map, with a comparable or better resolution, produced in a similar time span as the TDM WBL map. Usually, such maps are only available at local scale. To perform such a task, we have utilized the

Copernicus HRL WAW digital map (described in Section 2.2). Additionally, an accuracy comparison with the TDM WAM is included. The main scope of this additional analysis is twofold: on the one hand, we aim to provide the reader with a solid background and motivation that brought us to the development of the TDM WBL; on the other hand, we demonstrate the added value of such a new product, with respect to the state-of-the-art global water mapping product from TanDEM-X. Finally, a global-scale intercomparison with other global water maps is here performed, as well, by utilizing available products with a similar or coarser resolution, and generated with older input data. For this purpose, we compared the TDM WBL with the ESA CCI, FROM-GLC, and JRC GSW water maps. Such global products are generally less accurate than local-scale maps; nevertheless, they can give a good indication on the global performance.

### 4.2.1. TDM WBL Validation

The result of the validation using the Copernicus HRL WAW comprises overall 902 TDM WBL geocells over Europe. An overall accuracy of 99.1% has been obtained. However, the overall accuracy shows optimistic results, especially on imbalanced data sets, as most of the pixels under analysis are classified as non-water classes. More suitable parameters for the estimation of the accuracy on imbalanced data sets are the F-score and MCC parameters. In this case, a F-score of 83.2% and MCC of 81.4% have been obtained. Figure 9 shows the distribution of the mean F-score per geocell over the entire continent. Coastal areas as well as regions with high water content are well detected and mostly depicted in green, showing a F-score higher than 80% Inland regions with low water content often show a F-score lower than 50% and are indicated in red, orange, and yellow. The lower performance is mainly caused by an approximate detection of narrow rivers, as well as small lakes, due to the medium resolution (50 m) of TanDEM-X quicklook data. Moreover, some geocells with low F-score are still present in Scandinavia. This aspect mainly involves geocells, where only winter acquisitions were available and frozen water bodies were not completely detected. Note that 69.1% of the geocells have a water content higher than 1%. By discarding those geocells with less than 1% water content, an F-score of 93.0% and MCC of 90.1% are obtained, respectively, indicating the high quality of the produced map when non-negligible water bodies are actually present in the geocell. Figure 10 shows the confusion matrix for the validation of a single TDM WBL geocell, located in northern Italy. As it can be seen, most inconsistencies are found in correspondence of narrow rivers, while major water bodies appear well detected.

### 4.2.2. Comparison with TDM WAM

To assess the accuracy of the TDM WAM and compare it with the TDM WBL, we followed the same approach as in the previous Section 4.2.1, using as reference data the Copernicus HRL WAW binary map over Europe. Such a performance assessment approach requires, therefore, the comparison of binary water/non-water layers.

Figure 11 shows the difference in accuracy $\Delta_A$ between the TDM WBL and the TDM WAM over Europe. For each geocell, it has been computed as:

$$\Delta_A = F_{score}^{WBL} - F_{score}^{WAM}, \tag{9}$$

where $F_{score}^{WBL}$ and $F_{score}^{WAM}$ represent the mean F-score of the TDM WBL and the WAM, respectively, evaluated with respect to the Copernicus WAW map over Europe. In this case, the considered WAM binary layer includes the amplitude, as well as the coherence information (Coh + Amp in Table 1).

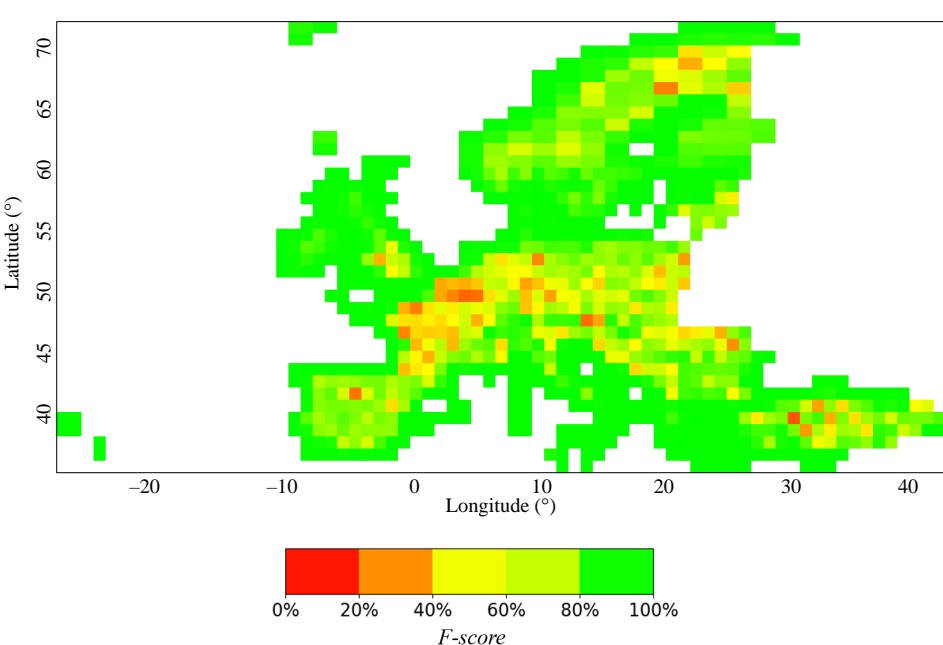

**Figure 9.** F-score on a geocell basis obtained for the 902 geocells used for the validation of the TDM WBL over Europe. The Copernicus HRL WAW has been used as reference map.

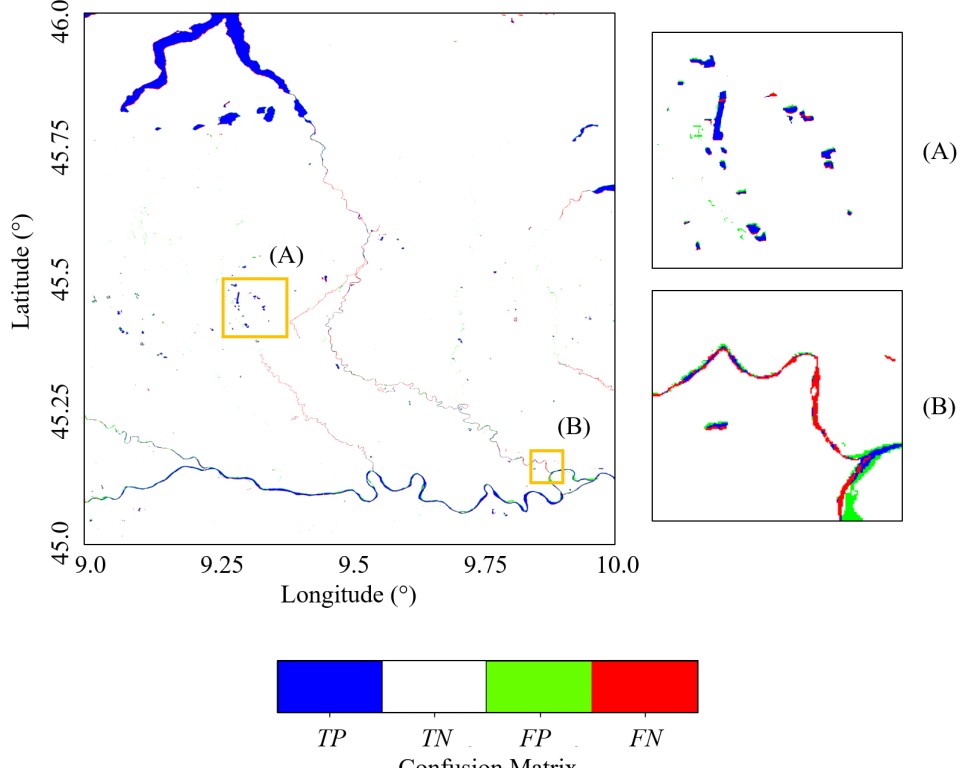

**Figure 10.** Details of the validation map (confusion matrix) for a geocell in Northern Italy. The TDM WBL has been compared with the Copernicus HRL WAW layer. Blue indicates water detected in both maps (*TP*), red indicates water detected only in the reference map (*FN*), green indicates water detected only in the TDM WBL map (*FP*), and white indicates no water detected in both maps (*TN*). The areas (A,B) highlighted in orange are depicted on the right-hand side.

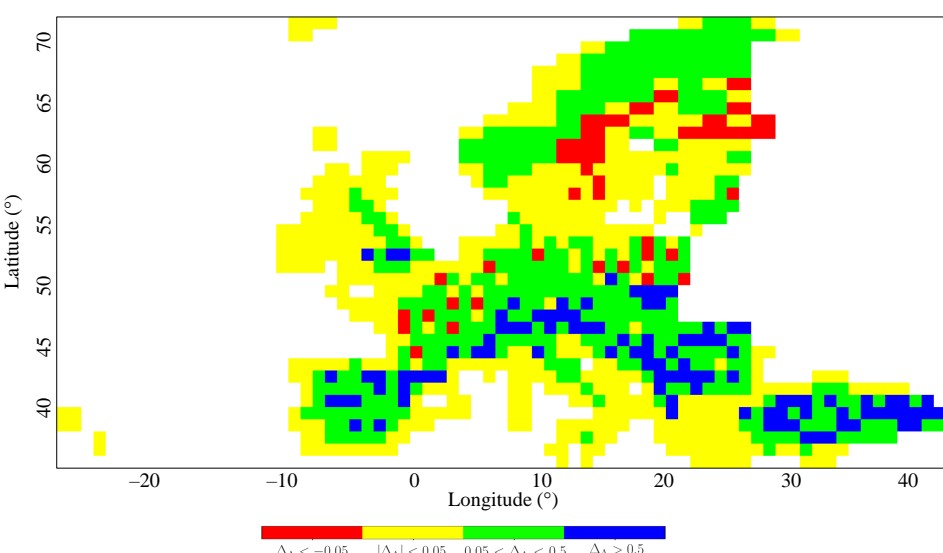

**Figure 11.** Accuracy comparison of the (F-score) on a geocell basis between the TDM WBL and TDM WAM over Europe. The Copernicus WAW layer is used as reference. Red: WAM better than WBL ($\Delta_A < -0.05$); yellow: comparable performance ($|\Delta_A| < 0.05$); green: WBL bettern than WAM ($0.05 < \Delta_A < 0.5$); blue: WBL significantly better than WAM ($\Delta_A > 0.5$).

　　Among the 902 geocells considered over Europe, for 39 tiles, the TDM WAM shows better performance than the TDM WBL, which are shown in red, and mainly correspond to regions where only winter acquisitions are available. In this case, the use of the amplitude, in addition to the coherence, helps improving the classification. Moreover, the difference in resolution of the input data, when generating the WBL and WAM maps seems to affect the performance of the TDM WBL over some disperse single inland geocells, where the water content is lower than 1%, as well. A total of 436 geocells are colored in yellow, covering mostly the European coastal areas, where both maps show a similar performance in the detection of open water. Overall, 427 geocells are depicted either in green or in blue, confirming the significant improvement of the WBL, with respect to the WAM.

　　In Table 4, the mean F-score value obtained over Europe for the different WAM water maps is indicated and compared with the one obtained for the TDM WBL and temporary and permanent WBL. By considering the 902 geocells, both WBL maps achieve an F-score of 83%, while the WAM considering both, coherence and amplitude, reaches only a 67%. From the different WAM combinations, different F-score values are obtained, spanning from 55% up to 81%, which make the a priori selection of the right combination difficult.

　　We then further analyzed the accuracy of the WAM, by considering two exemplary test regions: the Alps (latitude in [44°N, 48°N], longitude in [6°E, 18°E]), and Scandinavia (latitude in [58°N, 70°N], longitude in [4°E, 28°E]). The first is mainly characterized by the presence of small lakes and rivers in between the mountains, while the second is characterized by high-relief terrain on the west coast and the presence of frozen inland water bodies during winter.

　　In the case of the Alps region, the WBL (weighted mosaic) achieves a F-score of 77%, while the best WAM combination achieves 74%. Most of the misclassified pixels in the WAM correspond to areas of shadow and layover. These results confirm that the TDM WBL outperforms the WAM over high-relief terrain, demonstrating the importance of using a reliable shadow and layover mask, as the one exploited by the TDM WBL, for the detection of water bodies over mountainous regions.

**Table 4.** Comparison of the accuracy assessment (F-score index) on a geocell basis between the TDM WBL and the WAM over Europe (top) and, more specifically, the Alps (middle) and Scandinavia (bottom). For each test site, the first two rows identify the TDM WBL layers. Weighted refers to the TDM WBL (weighted mosaic), while Temp + Perm refers to the temporary and permanent water layer. Please refer to Table 1 for the explanation of the WAM combinations.

| | Water Map | Type | All Counters | Acq. Counter | | |
| | | | | 3 | 2 | 1 |
|---|---|---|---|---|---|---|
| Europe (902 geocells) | WBL | Weighted | 83.16 | - | - | - |
| | | Temp + Perm | 83.03 | - | - | - |
| | WAM | Coh + Amp | 67.16 | - | - | - |
| | | Coherence | 78.31 | 81.58 | 71.27 | 55.62 |
| | | Amp. <−15 dB | 69.57 | 65.46 | 72.38 | 61.29 |
| | | Amp. <−18 dB | 81.04 | 69.68 | 77.20 | 68.44 |
| Alps (48 geocells) | WBL | Weighted | 77.54 | - | - | - |
| | | Temp + Perm | 74.32 | - | - | - |
| | WAM | Coh + Amp | 40.68 | - | - | - |
| | | Coherence | 50.27 | 69.22 | 33.41 | 16.59 |
| | | Amp. <−15 dB | 45.45 | 66.05 | 50.80 | 31.63 |
| | | Amp. <−18 dB | 62.56 | 74.62 | 60.26 | 38.81 |
| Scandinavia (143 geoc.) | WBL | Weighted | 84.63 | - | - | - |
| | | Temp + Perm | 90.91 | - | - | - |
| | WAM | Coh + Amp | 77.02 | - | - | - |
| | | Coherence | 86.97 | 66.56 | 77.97 | 78.36 |
| | | Amp. <−15 dB | 78.66 | 57.15 | 74.68 | 70.67 |
| | | Amp. <−18 dB | 84.90 | 57.78 | 79.87 | 79.35 |

In the case of Scandinavia, the best performance of the WAM water maps are obtained for the coherence only and the strict threshold on the amplitude, by considering in both cases water detected in 1, 2, and up to 3 or more acquisitions (column "All counters" in Table 4). Here, an F-score of 86.97% and 84.90% is achieved, respectively. The effect of frozen water bodies is clearly visible in the WBL, which achieves a lower performance than the one obtained by considering the temporary and permanent WBL, which reaches a F-score higher than 90%.

The variable performance achieved by the different WAM combinations, as well as the improvement obtained by the TDM WBL, are clearly visible by looking at the confusion matrices of Scandinavia for different WBL and WAM layers, evaluated with respect to the Copernicus WAW reference map and depicted in Figure 12.

4.2.3. Intercomparison with Global Water Maps

The TDM WBL has been compared on a geocell basis, with other avaliable global water products: the ESA CCI water map, FROM-GLC water layer, and JRC GSW map. Even though such global products do not represent highly reliable reference data, such as the Copernicus HRL over Europe, they allow for a first assessment of the TDM WBL performance on a global scale. Table 5 summarizes the results obtained when considering

only geocells with a water content higher than 1%. In most of the regions, a F-score value, as well as a *MCC* value higher than 85%, are achieved, while the *OA* is well above 90%. Only when considering permafrost regions in the Northern Hemisphere a lower performance is obtained. This is clearly visible in Figure 13, which depicts the mean F-score obtained from the comparison of the global TDM WBL with the ESA CCI water map on a geocell basis. Geocells with less than 1% water content are excluded from the analysis and indicated in white. The TDM WBL exhibits a very good agreement with the reference map along coastal areas and in regions with high water content, such as the Amazon River basin, Patagonia in South America, and the islands in the Caribbean sea, as well as in the Indian ocean, Japan, Mediterranean sea, and northern Europe. A lower performance is mainly visible in the regions surrounding the Arctic ocean. This is due to the presence of frozen water surfaces in the mosaicked winter acquisitions. These effects are significantly mitigated when using the associated temporary and permanent water layer, as already demonstrated in Section 4.2.2. Overall, the obtained results confirm the good agreement of the TDM WBL with other global water products, making it a reliable reference on a global scale.

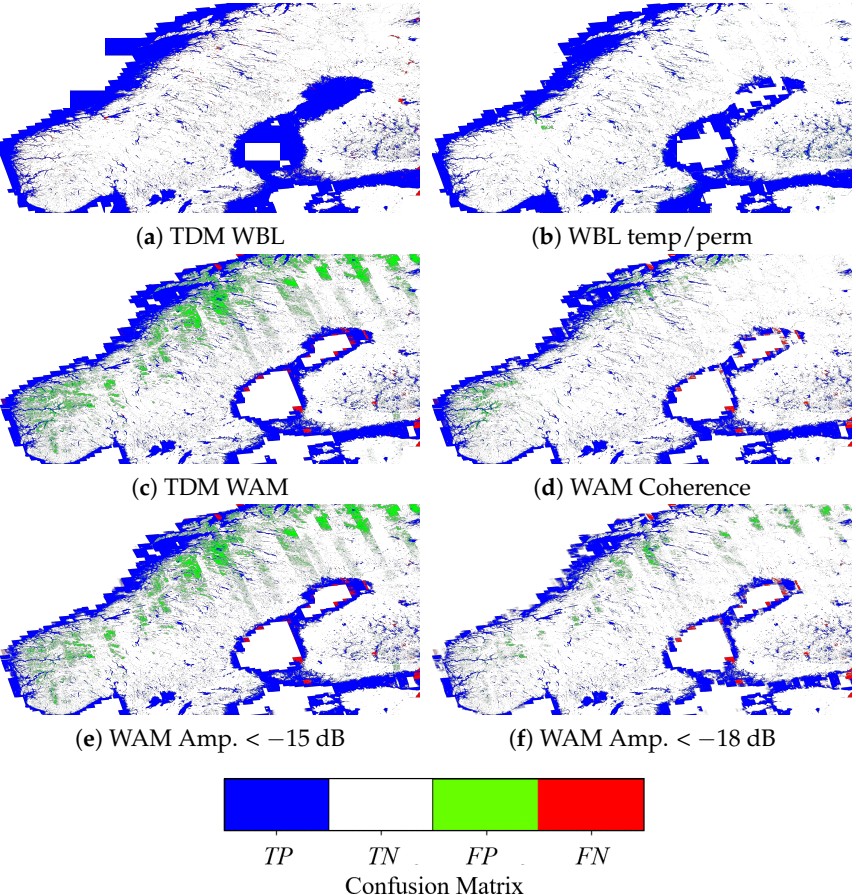

**Figure 12.** Confusion matrices between the TDM WBL and TDM WAM binary layers over Scandinavia. The Copernicus WAW layer has been used as reference. (**a**) Weighted WBL mosaic (TDM WBL), (**b**) temporary and permanent WBL layer (WBL temp/perm). The different versions of the TDM WAM correspond to (**c**) WAM coherence and amplitude, (**d**) WAM coherence only, (**e**,**f**) WAM amplitude with different thresholds. All TDM WAM versions have been generated considering all acquisitions counters, as indicated in Tables 1 and 4.

**Table 5.** TDM WBL comparison results with global water maps: the ESA CCI water map (CCI), FROM-GLC map (FGLC), and JRC GSW map (GSW). Only geocells with a water content higher than 1% are considered.

| Region of Interest | Nr. Geocells | | | OA | | | F-Score | | | MCC | | |
|---|---|---|---|---|---|---|---|---|---|---|---|---|
| | CCI | FGLC | GSW | CCI | FGLC | GSW | CCI | FGLC | GSW | CCI | FGLC | GSW |
| Canada | 1386 | 1366 | 1370 | 94.62 | 94.15 | 95.13 | 74.31 | 74.71 | 73.65 | 72.53 | 71.12 | 71.46 |
| USA and Mexico | 927 | 893 | 878 | 98.55 | 98.78 | 98.67 | 86.90 | 89.04 | 90.22 | 84.09 | 86.67 | 87.25 |
| Central and South America | 999 | 823 | 892 | 98.66 | 98.65 | 98.01 | 87.70 | 86.60 | 90.52 | 82.49 | 84.65 | 85.08 |
| Europe | 1080 | 1042 | 1014 | 98.42 | 98.64 | 98.52 | 86.81 | 88.78 | 90.14 | 85.17 | 87.23 | 88.50 |
| Africa | 791 | 717 | 689 | 99.17 | 99.15 | 99.05 | 90.85 | 91.43 | 95.27 | 88.24 | 90.08 | 92.91 |
| Asia | 2288 | 2162 | 2033 | 95.76 | 95.62 | 95.37 | 68.97 | 70.03 | 73.24 | 67.18 | 68.44 | 70.85 |
| Oceania | 997 | 840 | 903 | 99.08 | 99.21 | 98.32 | 94.24 | 95.72 | 97.25 | 88.66 | 90.90 | 90.62 |
| Greenland | 252 | 222 | 161 | 95.21 | 82.58 | 77.05 | 80.01 | 58.38 | 78.19 | 74.54 | 42.53 | 37.24 |

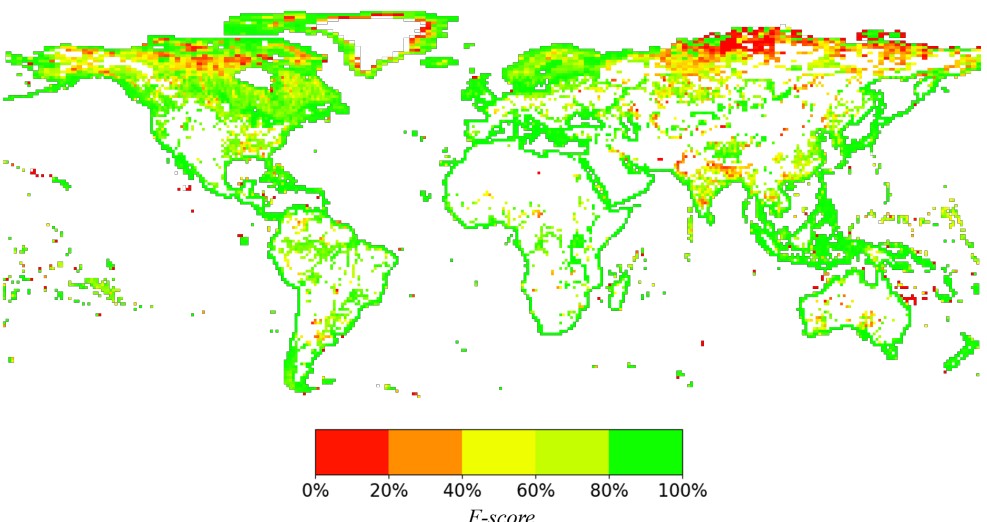

**Figure 13.** Mean F-score per geocell, obtained from the intercomparison of the TDM WBL, with the ESA CCI water map. Land regions with less than 1% water content are depicted in white and excluded from the analysis.

## 5. Discussion

The results obtained through the validation and the intercomparison activities demonstrate the high potential of the TanDEM-X interferometric data set to generate global products and the suitability of the proposed algorithms for extracting the location and extension of water surfaces from such a data set. The comparison of the generated TDM WBL map with other reference maps shows a strong agreement, when considering geocells with a water content higher than 1%, for which the performance evaluation is meaningful. The classification performance could be further improved by exploiting the TanDEM-X full-resolution data, set at 12 m × 12 m, which would allow for a better detection of narrow river beds and small lakes. In this way, it would also be possible to avoid the use of OSM data to properly set the water seeds over such regions.

From the validation and intercomparison activities, one can also assess that the implemented approach is robust and delivers a consistent and homogeneous data set at global scale. It is worth noting that water is a highly changing environment, which may respond differently to the interferometric radar system, depending on its status. For example, irrigated areas or wetlands appear as low coherent areas only when water is present on

the surface. Such a changing behavior makes the correct detection of water bodies from remote sensing systems at global scale an extremely challenging task.

The proposed two-step mosaicking strategy with the definition of the reliability weights shows the importance of correctly handling the input data. In the case of the TanDEM-X mission, the acquisitions planning has been focused on the generation of a high-performance DEM. For this objective, height of ambiguities in the order of 50 m during the first year of acquisitions (2011–2012) and around 35 m during the second year (2013–2014), were selected. Small height of ambiguities reduce the height error and improve the quality of the DEM [32]. However, for classification purposes, based on the interferometric coherence, such small values of the height of ambiguity may cause misclassification specially over forested areas [36]. In such images, acquired with very low values of height of ambiguities, both forests and water show very low coherence, and it is necessary to carefully consider this aspect. In areas where only such data takes are available, misclassification may occur, reducing the performance of the generated TDM WBL.

Beside the height of ambiguity, other aspects to be taken into account when planning the data takes are the acquisition geometry and the acquisition date. TanDEM-X offers a high acquisitions versatility by combining right- and left-looking modes with ascending and descending orbit directions. The operational satellites viewing direction is right-looking with respect to the flight direction. By combining acquisitions in ascending and descending orbits, it is possible to illuminate the same point on the Earth from opposite viewing angles. Such an approach has been exploited especially over mountainous regions to reduce missing data due to shadow and layover effects. In the case of water classification, when water bodies are surrounded by steep terrains, such as lakes surrounded by mountains or cliffs, the availability of images acquired from different points of view can significantly improve the final classification [40].

Seasonality also represents a key-factor for correctly detecting water from InSAR acquisitions. Water bodies in winter might be misclassified as non-water if the surface is frozen and therefore characterized by higher coherence. In order to understand the challenge of generating a global water body layer from the TanDEM-X data set, one can consider Figure 14, which shows the percentage of TanDEM-X nominal acquisitions acquired in winter in the Northern Hemisphere for latitudes above 50° (North America in Figure 14a, Europe Figure 14b, and Asia Figure 14c). Green indicates that all scenes have been acquired in summer, and red that all scenes have been acquired in winter time. As it can be seen, over many areas winter acquisition only are available, where most of water surfaces are frozen. This aspect is confirmed by the correlation between Figures 14 and 13. Indeed, for latitudes above 70°, regions acquired only in winter achieve significantly lower F-score values. This strongly impairs the final product accuracy and for this reason we developed a dedicated detection of inland frozen water (Section 3.5.1). On the contrary, at coastal areas, it has been possible to accurately detect the shoreline in the presence of summer data takes only.

In addition to all the previously mentioned aspects, the comparison of water maps acquired at different times and with different resolutions has to be carefully addressed as well. Indeed, higher resolution images can capture more details, while temporal changes in the extension of water surfaces inevitably leads to a lower agreement among the compared maps. Such changes can be caused by natural erosion, extreme hazards, as well as human activities.

Nevertheless, if, on the one hand, the time difference between water maps to be compared significantly affects the resulting performance, on the other hand, it also represents a key asset for detecting temporal changes occurring on the Earth's surface, as done, e.g., for the monitoring of wetlands [59,60].

An example of temporal changes between the TDM WBL and the ESA CCI water map is presented in Figure 15. Here, modifications in the river bed of the Amazon river can be identified through the analysis of the confusion matrix. Indeed, false positives (*FP*, green)

and false negatives (*FN*, red) can be associated to temporal changes in the water extent, corresponding to newly flooded areas and to dried ones, respectively.

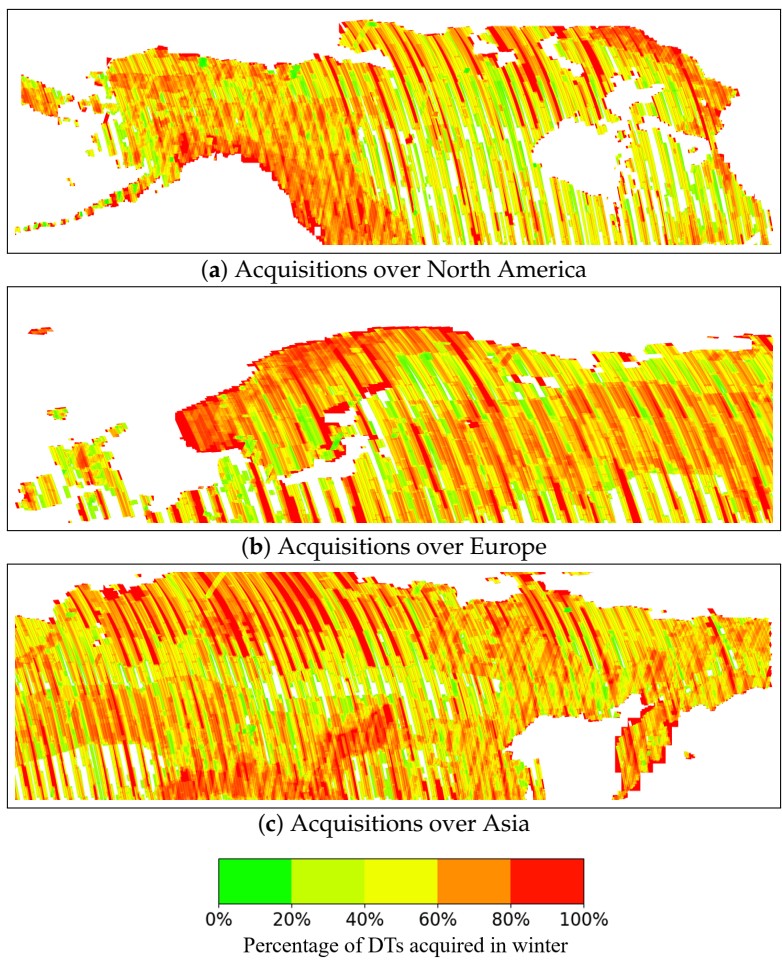

(**a**) Acquisitions over North America

(**b**) Acquisitions over Europe

(**c**) Acquisitions over Asia

0%    20%    40%    60%    80%    100%
Percentage of DTs acquired in winter

**Figure 14.** Percentage of acquisitions acquired in winter on the Northern Hemisphere, for latitudes between 50°N and 75°N. Winter is defined from October to March.

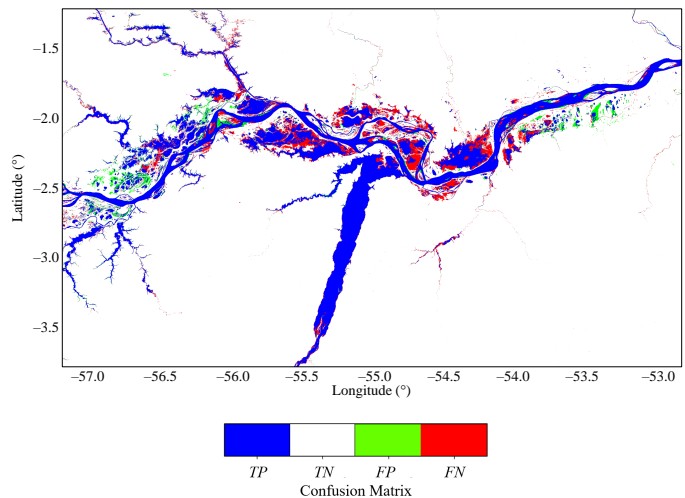

TP    TN    FP    FN
Confusion Matrix

**Figure 15.** Temporal changes detected between the TDM WBL and the ESA CCI water map in a time span of more than three years for the Amazon river, Brazil. Blue: stable water surfaces (*TP*), white: stable non-water areas (*TN*), green: newly flooded areas (*FP*), red: dried water surfaces (*FN*).

## 6. Conclusions

In this work, we presented the new global water body layer, generated from the TanDEM-X InSAR data set: the TDM WBL. The paper provides the end-users with a complete description of the algorithms and product peculiarities and performance. More than half a million bistatic scenes, covering all the Earth's landmasses, have been acquired and processed, since the beginning of the TanDEM-X mission. As in the case of the TanDEM-X forest/non-forest map, we have used an averaged and downsampled version of the original full-resolution data at a ground independent pixel spacing of 50 m × 50 m, which represents a good compromise between final product resolution and resulting computational burden.

The low interferometric coherence, which characterizes water bodies in InSAR data, has been exploited as input observable to a watershed-based classification algorithm. The proposed classification approach and two-step mosaicking strategy, which aims at an optimum combination of the multi-temporal and multi-baseline TanDEM-X data set, have been presented, as well.

The mosaicked product has been validated and compared with existing water classification maps, achieving excellent performance, with F-score index typically above 90% for geocells with a water content higher than 1%. Moreover, it represents a significant improvement, with respect to the current WAM as by-product of the TanDEM-X global DEM, since it provides a more homogeneous, ready-to-use binary product with a higher quality. This is particularly visible over difficult terrain areas, such as mountainous and vegetated regions.

The global TanDEM-X WBL, including the temporary and permanent water layer presented in this paper, will be open to the scientific community for free download and usage.

**Author Contributions:** J.-L.B.-B., M.M., F.S. and P.R. conceived and designed the methodology and the experiments; J.-L.B.-B. and P.V. performed the experiments; J.-L.B.-B., C.G. and P.R. analyzed the data; J.-L.B.-B., P.V., P.P., F.S., C.G. and A.P. contributed to the materials and analysis tools; J.-L.B.-B., M.M. and P.R. wrote and reviewed the paper; P.R. supervised the work. All authors have read and agreed to this version of the manuscript.

**Funding:** This research received no external funding.

**Data Availability Statement:** Openstreet Map data copyrighted OpenStreetMap contributors and available from https://www.openstreetmap.org (accessed on 20 May 2021).

**Acknowledgments:** The authors would like to thank the collaboration of our colleagues A. Pulella and F. Sica in the earlier stages of this study.

**Conflicts of Interest:** The authors declare no conflict of interest.

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
