# Peer review of "The Global Water Body Layer from TanDEM-X Interferometric SAR Data"

_remotesensing, doi:10.3390/rs13245069_

Round 1

Reviewer 1 Report

This is an excellent paper. It uses TanDEM-X InSAR data to create a global map of water bodies. The methodology is clearly defined and explained. The assessment of results is very clear. The paper refers to other similar data bases and is very sophisticated in its use of complementary data sets. The paper has extensive references and clear justifications.

Author Response

Dear Reviewer,

We thank you very much for your very positive comments and to endorse the paper for publication.

Sincerely,

The Authors.

Reviewer 2 Report

I appreciate the contribution of the results that this research will provide to the community in the form of an open global water dataset. The research is of practical significance to the community. I also appreciate the authors dealing with the spatio temporal dynamics of water bodies, which was addressed in the procedure and the outputs.  
I do not have any objections or comments before publication. 

The paragraph starting in page 2 line 65 could be made clearer and less ambiguous 

SARinterferometry(InSAR) allows for overcoming the above.....

There is only a few references to the width of the rivers that are detected by the mosaic. This is an important topic to address. Wich a cell size of 50m, the larger share of rivers and detail would be lost.  Could this be solved by a further development where the full resolution of the images is exploited? 

Author Response

Dear Reviewer,

We thank you very much for your comments, which for sure will help to improve the quality of the paper.

Please, find attached our detailed answers.

Sincerely,

The Authors.

Reviewer 3 Report

The present manuscript proposes a new water-bodies product derived from TDX.

The topic is of great interest, and the paper is well written and results nicely presented. The authors show that this new products outperforms the WAM layer. This is supported by a comprehensive presentation of the experiments and results.

In my opinion the manuscript is ready to be published as it is.

Author Response

Dear Reviewer,

We thank you very much for your review and for endorsing the paper for publication.

Sincerely,

The Authors.

Reviewer 4 Report

This paper uses bistatic interferometric coherence as the primary input feature of water detection and classifies the water surface in a single TanDEM-X image based on the watershed segmentation algorithm, and finally obtains a new global water body layer, the product generated by this method is compared with the existing water classification maps and has achieved excellent performance. This is an interesting research paper. There are some suggestion for revision.

  1. The motivation of the proposed solution is not clear. Please specify the importance of the proposed solution.
  2. Please highlight the innovations of the proposed solution in introduction.
  3. Most of references are a little bit out of date. Please discuss more recently published solutions, such as "Atmospheric Light Estimation Based Remote Sensing Image Dehazing", Remote Sensing 13 (13), 2432, 2021 and "Remote sensing image defogging networks based on dual self-attention boost residual octave convolution", Remote Sensing 13 (16), 3104, 2021.
  4. It is recommended to introduce the differences and advantages of the proposed solution and other existing methods in introduction, as well as the specific issues that this article solves.
  5. It is recommended to add a visual display of the corresponding data in the subsection "The TanDEM-X Interferometric Global Data Set".
  6. Does seasonally frozen lake data require manual screening?
  7. Deep learning has achieved good results in various fields of research. Do you consider using deep learning image segmentation methods in single-scene water classification?
  8. Figure 4 does not provide an intuitive analysis. It is recommended to add explanatory marks in the figure to assist the preview.
  9. What is the implementation process of the watershed segmentation algorithm? It is recommended to introduce in detail.
  10. When mosaicking regions at latitudes higher than 60°N, an ad-hoc approach is used to deal with frozen water surface detection. Has the frozen water surface in other areas been processed? If yes, please specify how it works.
  11. The experimental results are not convincing. Please compare the proposed solution with more recently published solutions.
  12. Please specify how to obtain the suitable weights in different datasets.

Author Response

Dear Reviewer,

We thank you very much for your comments. Please, find attached our answers.

Sincerely,

The Authors.

Round 2

Reviewer 4 Report

All my concerns have been addressed. I recommend this paper for publication.